# Dynamic Projections of Extreme Sea Levels for western Europe based on Ocean and Wind-wave Modelling

Alisée A. Chaigneau[1,2†*], Angélique Melet[2], Aurore Voldoire[1], Maialen Irazoqui Apecechea[2], Guillaume Reffray[2], Stéphane Law-Chune[2] and Lotfi Aouf[3]

[1] CNRM, Université de Toulouse, Météo-France, CNRS, Toulouse, France.
[2] Mercator Ocean International, Toulouse, France.
[3] Météo-France, Toulouse, France.

[†] now at IHCantabria - Instituto de Hidráulica Ambiental de la Universidad de Cantabria, Santander, Spain.

*Correspondence to*: Alisée A. Chaigneau (alisee.chaigneau@gmail.com)

**Abstract.** Extreme sea levels (ESLs) are a major threat for low-lying coastal zones. Climate change induced sea level rise (SLR) will increase the frequency of ESLs. In this study, ocean and wind-wave regional simulations are used to produce dynamic projections of ESLs along the western European coastlines. Through a consistent modelling approach, the different contributions to ESLs such as tides, storm surges, waves, and regionalized mean SLR are included as well as most of their non-linear interactions. This study aims at assessing the impact of dynamically simulating future changes in ESL drivers compared to a static approach that does not consider the impact of climate change on ESL distribution. Projected changes in ESLs are analysed using non-stationary extreme value analyses over the whole 1970-2100 period under the SSP5-8.5 and SSP1-2.6 scenarios. The impact of simulating dynamic changes in extremes is found statistically significant in the Mediterranean Sea with differences in the decennial return level of up to +20% compared to the static approach. This is attributed to the refined mean SLR simulated by the regional ocean general circulation model. In other parts of our region, we observed compensating projected changes between coastal ESL drivers, along with differences in timing among these drivers. This results in future changes in ESLs being primarily driven by mean SLR from the global climate model used as boundary conditions, with coastal contributions having a second order effect, in line with previous research.

## 1 Introduction

Coastal zones are among the most densely populated and urbanized areas in the world. 10% of the world's population lives in low-elevation coastal zones with 50 million people in Europe (McMichael et al., 2020; Neumann et al., 2015; Wolff et al., 2020). Coastal zones are also increasingly threatened by sea level rise (SLR) and the associated increase in frequency of extreme sea levels (ESLs), during which most damage occurs (e.g., Fox-Kemper et al., 2021; Le Cozannet et al., 2022). Without adaptation measures, the annual number of Europeans exposed to coastal flooding could reach 1.5-3.6 million by the end of the century and the associated expected annual damages could reach EUR 90-960 billion (Vousdoukas et al., 2018a).

Sea level varies over a range of time scales due to a combination of processes and their interactions (Woodworth et al., 2019; Idier et al., 2019). At the coast, sea level variations result from the superposition of global mean SLR, regional mean sea level changes, and local sea level changes. ESLs at the coast are primarily due to a combination of astronomical tides, storm surges (due to low atmospheric surface pressure and wind setup), and wind-waves.

At global and regional scales, projections of ESLs have mostly been analysed based on tide gauge data (Vitousek et al., 2017, Rasmussen et al., 2018; Lambert et al., 2020; Lowe et al., 2021; Rashid et al., 2021; Woodworth et al., 2021; Tebaldi et al., 2021; Rasmussen et al., 2022; Hermans et al., 2023). These studies employ a static approach, where the past

distribution of coastal sea level extremes (e.g., tides, surges) is simply shifted by projected mean relative SLR, assuming a statistical distribution not altered by climate change (Kirezci et al., 2020; Lambert et al., 2020; Almar et al., 2021). In this case, the quality of the analysis is limited by the length of the available historical time series. In addition, the static approach is mostly based on tide gauge records, which only partly capture wave contribution to ESLs (e.g., Woodworth et al. 2019).

Thanks to the use of numerical models, the different contributions can be simulated dynamically over the historical period and future climates. As dynamic approaches are computationally expensive, their use for regional to global projections of ESLs is recent (Fox-Kemper et al., 2021, Melet et al., 2024). They have been mostly applied with 2-D barotropic hydrodynamic models, forced by atmospheric fields simulated by climate models (Palmer et al., 2018; Vousdoukas et al., 2017, 2018, Jevrejeva et al., 2023) and potentially by accounting for future SLR (Muis et al., 2020, 2023). These studies emphasize that future changes in frequency of ESLs primarily depend on mean SLR rather than on changes in other components such as storm surges or tides (Vousdoukas et al., 2017, 2018b; Muis et al., 2020b; Jevrejeva et al., 2023), with the wave contribution often being omitted (Melet et al., 2024). However, recent studies have identified significant trends in various ESL drivers over past (Pineau-Guillou et al., 2021; Roustan et al., 2022 for the tides, Calafat et al., 2022; Tadesse et al., 2022 for the storm surges) and future periods (Haigh et al., 2019 for tides and Muis et al., 2023, Dullaart submitted for storm surges, Hemer et al., 2013; Aarnes et al., 2017; Meucci et al., 2020; Lobeto et al., 2021; Melet et al., 2020; Morim et al., 2021, 2023 for waves), suggesting the necessity of dynamic approaches. In addition, these studies based on a dynamic approach often omit non-linear interactions between ESL drivers notably between waves and sea level although they can be important, especially considering future SLR (Arns et al., 2017; Idier et al., 2019; Arns et al., 2020; Bonaduce et al., 2020; Staneva et al., 2021a; Chaigneau et al., 2023).

High-resolution 3-D ocean general circulation models such as NEMO can also be used for simulating dynamical changes in ESLs. These models can provide more consistent simulations by simulating changes in mean sea level (due to ocean circulations and addition of mass to the ocean), changes in storm surges and tides, but also the non-linear interactions between all these components. Additionally, they can also be coupled with wave models to account for the wave contribution (Lewis et al., 2019; Staneva et al., 2021b). Due to the high computational cost, their application in long-term ESLs studies has been extremely limited (Chaigneau et al., 2022), and their utilization in mean sea level projections typically focused on specific regions (e.g., Northern Atlantic and North Sea in Hermans et al., 2020, Chaigneau et al., 2022; Chinese Seas in Kim et al., 2021 and Jin et al., 2021; Sannino et al., 2022 for the Mediterranean Sea).

The aim of the present study is to assess the impact of dynamically simulating projected changes in ESLs using a consistent regional modelling approach for western European coasts. To do that, regional general ocean circulation and wind-wave simulations from Chaigneau et al. (2022) and Chaigneau et al. (2023) are used for the 1970-2100 period under the SSP5-8.5 and SSP1-2.6 climate change scenarios. These simulations include the different sea level contributions (mean sea level, tides, storm surges, waves) and their interactions. To our knowledge, this is the first time that such a regional baroclinic ocean modelling approach is used to assess the long-term changes in ESLs, including the wave contribution. Non-stationary extreme value analyses are applied to time series including all the different sea level components for the whole period. These analyses are compared to the static approach (historical distribution shifted by the mean SLR) to assess the importance of considering dynamic changes in ESLs. ESL projections are analysed in terms of changes in return levels (allowances) and return periods (amplifications) for the 1-in-10-year and 1-in-100-year events. However, this methodology is not fitted to provide local-scale projections of ESLs that would require local parameters to be considered, depending on each location or beach. The focus of this study is rather on identifying regional specific key processes or mechanisms that need to be considered in projections of ESLs. The paper is organized as follows. Regional ocean and wave simulations are presented in

Sect. 2 together with the extreme value analysis used to compute historical and future return levels for both static and dynamic approaches. Sect. 3 provides the regional validation of ESLs against tide gauge data. In Sect. 4, projected changes in ESLs under the SSP5-8.5 and SSP1-2.6 scenarios are presented. The impact of the dynamic approach on future changes in ESLs is evaluated, including for the wave contribution. Finally, results are discussed in Sect. 5, and conclusions are drawn in Sect. 6.

## 2 Data and Methods

### 2.1 Tide gauge data

The modelled historical ESLs are validated against GESLA3 (Global Extreme Sea Level Analysis GESLAv3) high-frequency (at least hourly) tide gauge records (Haigh et al., 2023, Woodworth et al., 2016). The validation period spans 45 years because the historical simulations cover the 1970-2014 period. Tide gauge stations with a temporal data coverage of at least 60% over the 1970-2014 period are selected. We therefore focused the validation on the 1-in-10-year level instead of the 1-in-100-year level, as the uncertainties associated with estimates of the 1-in-10-year return period are lower for such a period. Given the horizontal resolution of the regional models (Sect. 2.2), tide gauges located in specific locations such as estuaries, channels, and bays as in the Netherlands were discarded in this study.

### 2.2 Regional sea level simulations

Projected changes in ESLs are analysed along the north-eastern Atlantic coasts based on hourly outputs from consistent regional ocean and wave simulations. The domain covered by the regional simulations is called IBI for Iberian-Biscay-Ireland (Fig. 1). It extends from 25 to 65 °N and 21 °W to 14 °E and includes the north-eastern Atlantic Ocean, the North Sea, and the western Mediterranean Sea. This region presents a diverse range of physical processes relevant in modelling ESLs (Fig. 1). The English Channel and its adjacent Atlantic area are subject to significant sea level variations, primarily driven by tidal signals of up to 10 meters (Valiente et al., 2019; Stokes et al., 2021). The North Sea has a mesotidal regime and is characterized by strong winds from intense storms, leading to substantial storm surge events (Marcos and Woodworth, 2017). On the contrary, sea level variations in the western Mediterranean Sea are considerably smaller (Toomey et al., 2022), mainly due to its micro-tidal regime that rarely exceeds 50 cm. Regarding wave exposure, the Atlantic coast faces large swell events originating from the open ocean (Masselink et al., 2016; Bruciaferri et al., 2021), while both the North Sea and Mediterranean Sea are dominated by wind waves (wind sea) due to their protected location (Chen et al., 2002; Bergsma et al., 2022).

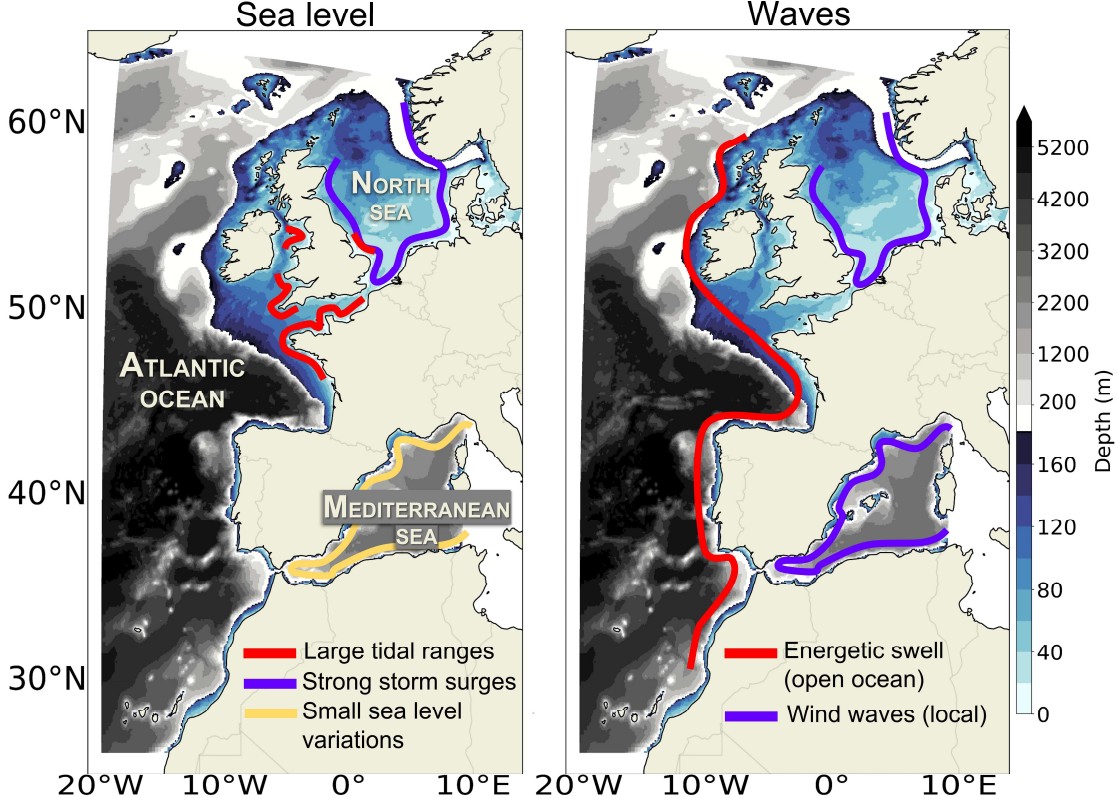

**Figure 1: Bathymetry (m) in the IBI region. The shelf-break defined by the 200-m isobath is indicated by the change in colour shades in the colormap. The dominant key processes contributing to ESLs are shown in colours for each part of the region.**

The regional ocean simulations are produced with a 3-D ocean circulation model at a 1/12° horizontal resolution (≈ 4–8.5 km for the latitudes of IBI) by a dynamical downscaling of a CMIP6 global climate model (GCM) at 1/4° spatial resolution for the ocean and ½° for the atmosphere (Saint-Martin et al., 2021; Voldoire et al., 2019). The regional wave simulations are produced at a 1/10° resolution (≈ 5.5-10 km for the latitudes of IBI) by a dynamical downscaling of a global wave model (1°), itself forced by the same GCM. Both the regional ocean and wave simulations are forced by the three-hourly winds of

the GCM. The particularity is that the wave model is also forced by the hourly sea level variations from the regional ocean model to include sea level-wave interactions that are important in the IBI domain due a large tidal range (Chaigneau et al., 2023). This consistent modelling approach provides ocean and wave simulations that are temporally phased. The simulations cover the 1970-2100 period under the high-emission, low-mitigation SSP5-8.5 and the low-emission, high-mitigation SSP1-2.6 scenarios. They are extensively described and validated in Chaigneau et al. (2022) for the ocean (mean sea level, general

circulation, water masses) and Chaigneau et al. (2023) for waves (mean and extreme significant wave height and peak period). Table 1 summarizes the different simulations used in the study.

| Name of the model | Model type | Name of the model | Historical time-span | Future time-span and scenarios | Horizontal resolution | Forcings | Application in the paper | Simulated sea level contribution | References |
|---|---|---|---|---|---|---|---|---|---|
| IBI-CCS | Regional 3-D ocean general circulation model | NEMO3.6 | 1970-2014 | 2015-2100 (SSP5-8.5, SSP1-2.6) | 1/12° | CNRM-CM6-1-HR (ocean and atmosphere variables) | Analyses | Regionalized mean sea level, tides, storm surges, interactions + thermosteric SLR added a posteriori | Chaigneau et al., 2022 |
| IBI-CCS-WAV | Regional wave model | MFWAM | 1970-2014 | 2015-2100 (SSP5-8.5, SSP1-2.6) | 1/10° | CNRM-CM6-1-HR (winds), IBI-CCS (surface currents, sea level), CNRM-HR-WAV (wave spectra) | Analyses | Wave contribution (modified by sea level variations) | Chaigneau et al., 2023 |
| CNRM-CM6-1-HR | Global climate model | NEMO3.6 (ocean), | 1970-2014 | 2015-2100 (SSP5-8.5, | 1/4° ocean | | Forcing | Mean sea level (dynamic sea level | Voldoire et al., 2019 |

| (GCM) | APEGE-Climat 6.3 (atm) | | SSP1-2.6) | ½° atm | | | and freshwater balance) + interactions + thermosteric SLR added a posteriori | Saint-Martin et al., 2021 |
|---|---|---|---|---|---|---|---|---|
| CNRM-HR-WAV | Global wave model | MFWAM | 1970-2014 | 2015-2100 (SSP5-8.5, SSP1-2.6) | 1° | CNRM-CM6-1-HR (winds, surface, currents, ice cover) | Forcing | | Chaigneau et al., 2023 |

**Table 2: List of the different simulations used in the study.**

## 2.3 Computation of total water level time series

Hourly outputs from regional ocean and wave simulations (Sect. 2.2) are combined to obtain the total water level (TWL) time series (1):

$$\eta_{TWL} = \eta_{SWL} + \eta_{wave} \quad (1)$$

The still water level (SWL) $\eta_{SWL}$ (eq. (1)) comes from the 3-D regional ocean simulations (Sect. 2.2, Tab. 1). The associated extreme events are called hereafter ESWLs (extreme still water levels). The SWL includes the contribution of regionalized mean sea level, tides, storm surges and non-linear interactions between all these processes. Here the mean sea level variations include changes 1) in the dynamic sea level due to ocean circulations and 2) in the mass variations due to the addition of water mass from cryosphere and land to the ocean, and to the balance between evaporation/precipitation/river runoff). The global thermosteric SLR is added to the SWL a posteriori as the regional ocean model relies on the Boussinesq hypothesis that does not allow the water column to expand (Griffies and Greatbatch, 2012).

The first order regional wave contribution $\eta_{wave}$ (eq. (1)) is evaluated using the wave simulations outputs (Sect. 2.2, Tab. 1) based on the generic parameterization of Stockdon et al. (2006) applicable for sandy beaches. The extreme events including this contribution are called hereafter ETWLs (extreme total water levels). The aim is not to represent the local behavior of waves but to consider a regional large-scale impact of waves on ESLs:

$$\eta_{wave} = 0.35\beta\sqrt{H_s L_p} \quad (2)$$

where $H_s$ is the deep-water significant wave height, $L_p$ is the wavelength related to the peak period $T_p$ through the deep-water linear dispersion relationship: $L_p = \frac{g}{2\pi}T_p{}^2$, $g$ is the acceleration of gravity, and $\beta$ is the foreshore beach slope. The foreshore beach slope is taken constant in space and time to 4 %. This value is representative of a global spatial-mean value found in a previous broad-scale study (Melet et al., 2020). A large-scale wave contribution scaling $\sqrt{H_s L_p}$ is also presented to allow our results to be scaled with different beach slopes or other empirical formulae (e.g., Melet et al., 2020). The limitations associated with this methodology are presented in the Discussion (Sect. 5).

## 2.4 Extreme value analyses

Due to a changing climate, sea level time series are expected to be non-stationary (i.e. statistical properties such as trend and variability that vary in time) and particularly due to long term SLR. Two types of approaches are used to derive ESLs in projections: the static approach based on historical data and the dynamic approach using both past and future information.

**Dynamic approach**

To consider long-term changes in ESLs, barotropic and baroclinic models can be used to simulate dynamical changes in the different contributions (mean sea level, tides, storm surges, waves and their non-linear interactions). To statistically analyse these long-term changes in simulating the different components, two methods are usually applied. The time slices method

has been used in Muis et al. (2020, 2023) and Mentaschi et al. (2016). This approach usually compares two 30-year past and future periods, assuming quasi-stationarity within each sub-period to which the stationary extreme value theory can be applied. However, the short duration of the slices poses challenges in confidently fitting extreme events, particularly for long return periods (e.g., 100-year return period). Another method is to use the full time series to assess the changes, which helps to reduce the confidence intervals for rare extremes such as the 1-in-100-year event. For instance, one approach is to fit non-stationary statistical models on the distribution parameters to make them time-dependent over the whole time period (Robin and Ribes, 2020). In this work, we use a method proposed by Mentaschi et al. (2016) and used in Vousdoukas et al. (2018b) and Mentaschi et al. (2017) that simplifies the former non-stationary method. The method uses predefined transformation functions to consider changes in ESL variability and trend for the whole simulated period.

The calculations are implemented on the 131-year time series (1970-2100) of $\eta_{SWL}$ and $\eta_{TWL}$ (Sect. 2.3). First, the principle is to transform the long-term non-stationary time series into a stationary series to which the stationary theory can be applied, with a time-constant estimate of the distribution parameters. Here, the extremes of the 131-year stationary transformed time series are locally fitted to a Generalized Pareto Distribution (GPD) with a peak over threshold method (following Wahl et al., 2017). A spatially variable exceedance threshold $u$ corresponding to an average of 3 events per year was chosen with an independence criterion of 3 days between two events for storm declustering (Wahl et al., 2017). Note that the selected extreme peaks over the threshold do not necessarily occur at the same time for $\eta_{SWL}$ and $\eta_{TWL}$. The GPD is specified by 3 parameters: $u$ the location parameter (corresponding to the threshold for selecting extremes), $\sigma$ the scale parameter ($\sigma > 0$) and $\xi$ the shape parameter controlling the tail of the distribution (Fig. 2a). The slope ($\sigma$) reflects the variability in the extremes as called in Lambert et al. (2020). This means that the steeper the curve, the larger the difference between rare extremes (i.e. the 1-in-100-year event) and more common ones (i.e. the 1-in-1-year event). The cumulative stationary distribution function $F$ of the GPD for x an extreme value selected above of the threshold $u$ is:

$$F_{u,\sigma,\xi}(x) = 1 - \left[1 + \xi\left(\frac{x-u}{\sigma}\right)\right]_+^{-\frac{1}{\xi}} (3),$$

for $\xi \neq 0$ and for $x \geq u$ if $\xi \geq 0$ and for $u \leq x \leq u - \sigma/\xi$ if $\xi < 0$

Then, to take into account the non-stationarity of the extreme value distribution, the GPD parameters $u$ and $\sigma$ are assumed to evolve in time, while the shape parameter $\xi$ remains constant (Marcos and Woodworth, 2017; Mentaschi et al., 2016). The parameters functions are chosen as:

$$u(t) = S(t) * u + T(t)$$
$$\sigma(t) = S(t) * \sigma \qquad (4)$$
$$\xi(t) = \xi = constant$$

with $t$ the time (in hours) from 1970 to 2100, $T(t)$ the long-term trend and $S(t)$ the long-term variability of the time series respectively computed as the running mean and running 99th percentile over a 20-year time-window (see Mentaschi et al., 2016 for more information on the stationary-transform extreme value analysis). Using these time evolving distribution parameters (representative of successive 20-year time periods), the profile of the return level curve changes over time, affecting the amplifications and allowances of future ESLs (Fig. 2b). The 95% confidence intervals associated with the extreme value analyses are also computed based on the whole 1970-2100 stationary time series and then made time dependent. The calculation of confidence intervals in the package used for this study (Mentaschi et al., 2016) relies on the Delta Method (asymptotic intervals) which tends to produce narrower and symmetric confidence intervals compared to other methods like the bootstrap method (Caires, 2011). This method has been used to propagate error components related to the

uncertainty in estimating the long-term trend and long-term variability (99th percentile) to the error associated with fitting

the stationary extreme value distribution, thereby combining both sources of uncertainty.

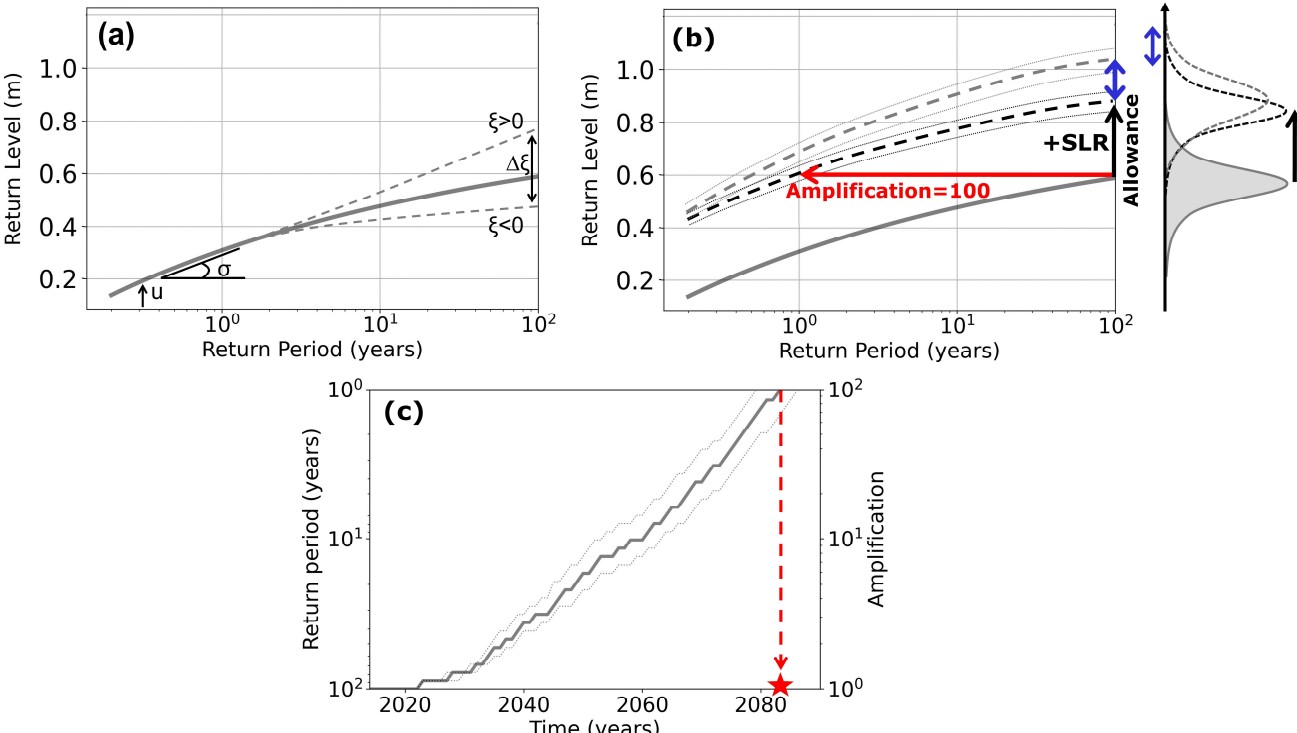

**Figure 2: Diagram with the different concepts used in the extreme event analyses. (a) Relation between ESLs (return levels, in m)
and associated return periods (in years). The distribution parameters are represented in black: u the location parameter**
**(threshold), σ the scale parameter and ξ the shape parameter for the past period (grey solid line). Graphically, the threshold for
the selection of extremes corresponds to the return level reached for a return period of 1/3 years as 3 events per year are selected.
(b) Same as a) but adding the curves for a future scenario (dashed lines), in black with the static approach which consists in adding
an offset to the return level corresponding to mean SLR (black arrows), in grey with a dynamic approach which consists in
considering also changes in the different coastal sea level contributions. The differences are also schematized with distributions on**
**the right side of panel (b). The blue arrows indicate the difference between both approaches. The difference is considered
significant when the confidence intervals (dotted lines) are disjoint. (c) Amplification of the past centennial event (HCE, 1-in-100-
year event) as a function of time. The red star highlights the year in which the HCE will become a yearly event i.e., when the
amplification factor reaches 100.**

**Static approach**

To assess the limitation of considering a static rather than a dynamic approach, we have also calculated projected changes in

ESLs using a static approach. For this purpose, we use the historical return period curves obtained for the 1995-2014 period

with the dynamic approach. As the long-term trend and variability of the dynamic approach are calculated over a 20-y time

window period, both are comparable. In projections, these historical curves obtained are shifted by mean regional SLR from

the GCM (Tab. 1) for the period 2081-2100. This is done for both $\eta_{SWL}$ and $\eta_{TWL}$. The differences using dynamic and static

approaches are illustrated with the blue arrow in Figure 2b. Using this method to calculate the static approach, dynamic

changes in extremes encompass changes in all simulated ESL components and interactions, as well as differences in the

mean SLR simulated by the regional ocean model and the GCM.

**2.5 Metrics used in the study**

Different metrics are used to analyse the ESLs and their projections. We focus on the past 1-in-100-year and 1-in-10-year

events defined as events that respectively have a 1% and 10% chance of exceedance in any given year. In projections, we

assess the allowances and amplifications of the 1-in-100-year and 1-in-10-year past events. The allowance is defined as the

change in amplitude of the ESLs (in meters) of a given probability extreme event and the amplification is the change in

frequency (in return periods) of a given threshold extreme event (Fig. 2b). Another metric used to analyse the projected

changes in ESLs is the year in the future when the past or historical centennial event (HCE = 1-in-100-year event over the past/historical baseline period) is expected to recur once a year on average, becoming an annual event (Fig. 2c). This corresponds to an amplification of 100 for the HCE.

## 3 Validation of the ESLs against tide gauge data

The modelled 1-in-10-year ESLs are validated below over the 1970-2014 period against tide gauge records (Sect. 2.1). The extreme value analysis method applied for the validation is the stationary theory (eq. (3)) for both tide gauge data and simulations, without accounting for waves, as they are only partly recorded in tide gauge data (Woodworth et al., 2019). Validation results for ETWL and for other return periods (1-in-5, 1-in-10, 1-in-20, 1-in-50 and 1-in-100-year events) are provided in Supplementary Table S4.1.

In the region, the highest values of decennial ESLs can reach more than 8 meters and are found in the macrotidal areas (Fig. 1a), including the Irish Sea, southern English Channel and Bristol Channel (Fig. 3a). In general, the errors at the different tide gauge stations rarely exceed 20% (Fig. 3b), which is consistent with values found in Muis et al. (2016, 2020) and Kirezci et al. (2020) for the region. Along the French Atlantic coast, the Mediterranean coasts and the northern Great Britain coasts, the modelled 1-in-10-year level is properly represented in comparison to available tide gauge data with biases less than 20 cm (Fig. 3b). In the eastern English Channel, Irish Sea, southern North Sea and Bristol Channel, the model underestimates the ESLs (Fig. 3b). The underestimation of ESLs is consistent with other studies such as Irazoqui Apecechea et al. (2023) and Kirezci et al. (2020). In Irazoqui Apecechea et al. (2023), a general underestimation of the extreme modelled storm surges along the North Sea coasts is also found. They related the ESLs underestimation to the too weak extreme winds in the models but also to the bathymetry that is not fine enough to correctly capture the ESL events in complex areas like the Netherlands. These two explanations are also valid in our case since we use forcing fields from a GCM with a resolution of 1/2° for the atmosphere and a regional ocean model at 1/12° (Tab. 1). Moreover, the regional ocean model does not allow for a very fine bathymetry representation and does not yet use the "wetting and drying" parameterization (O'Dea et al., 2020) that allows modelling of uncovered banks.

In conclusion, the modelled ESLs appear to be correctly represented compared to tide gauge data across different return periods (Tab. S4.1). The ESLs are however slightly underestimated as it is generally the case in the model-based studies at large scale.

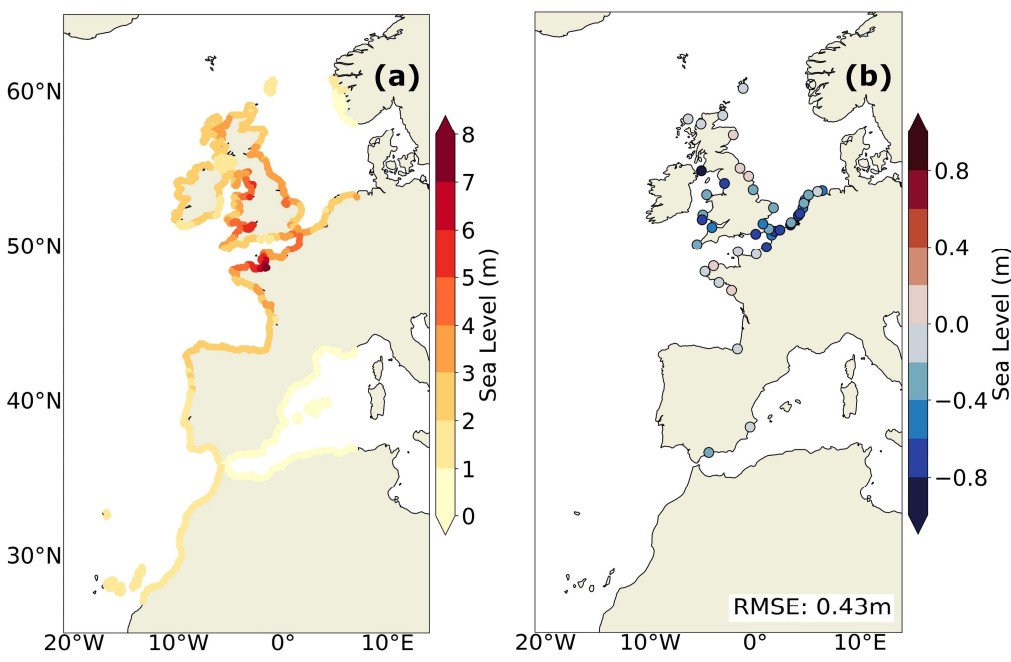

**Figure 3: (a) Modelled 1-in-10-year ESWL (in m) for the 1970-2014 period. (b) Bias between the modelled 1-in-10-year level and tide gauges from the GESLA3 dataset for the 1970-2014 period. The RMSE is calculated as the root mean squared deviations between modelled 1-in-10-year level and tide gauges.**

## 4 Dynamic projected changes in ESLs

### 4.1 Regionalized projected changes in ESLs

Future evolution of the HCE including changes in all the different sea level components is assessed under both scenarios (Fig. 4). A strong north-south gradient of the amplifications is observed, which is consistent with findings at global scale (Fox-Kemper et al., 2021; IPCC, 2019; Oppenheimer et al., 2019; Vousdoukas et al., 2018; Jevrejeva et al., 2023). This indicates that our single-forcing GCM is not an outlier compared to other GCMs. Differences of up to 40 years are observed between the two scenarios (Fig. 5a), regardless of the north-south gradient of the amplifications. South of 45°N, very strong amplifications are projected (Fig. 4). The most impacted zone are the Balearic and Canary Islands where the HCE is expected to become an annual event within 20 years (before 2045) with a small impact of the scenario considered (Fig. 5a). This phenomenon occurs because these southern regions are subject to a low variability in extremes (flat curves, negative shape parameter, Fig. 2a). In consequence, even a slight increase in sea level leads to large amplifications (Fig. 2b). In the north of the domain, HCEs will become annual events later, towards the end of the century, or after, for example, in the southern North Sea. These regions are prone to intense storm surge events, resulting in a high variability of extremes (steep curves, positive shape parameter, Fig. 2a). This variability typically leads to lower amplifications. Results including the wave contribution are provided in the Supplement Materials (Sect. S1 and S2), using sensitivity analyses for beach slopes (Sect. S3). As previously highlighted in Lambert et al. 2020, we found that including wave contribution delays by up to 30 years the HCE becoming annual along the European coasts. This is due to an increased variability in the extremes when accounting for waves because extreme events of waves and other sea level components do not occur at the same time.

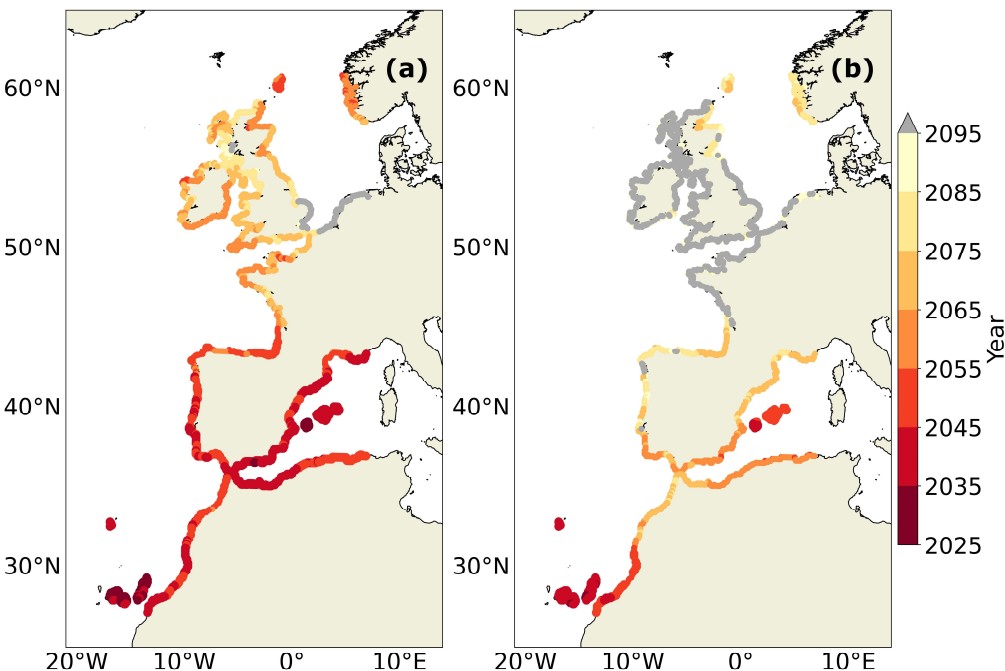

**Figure 4: (a) Year in which the HCE will occur once a year in the future under the SSP5-8.5 scenario for the SWL. The grey dots indicate the locations where HCEs do not recur annually before 2095. (b) same under the SSP1-2.6 scenario.**

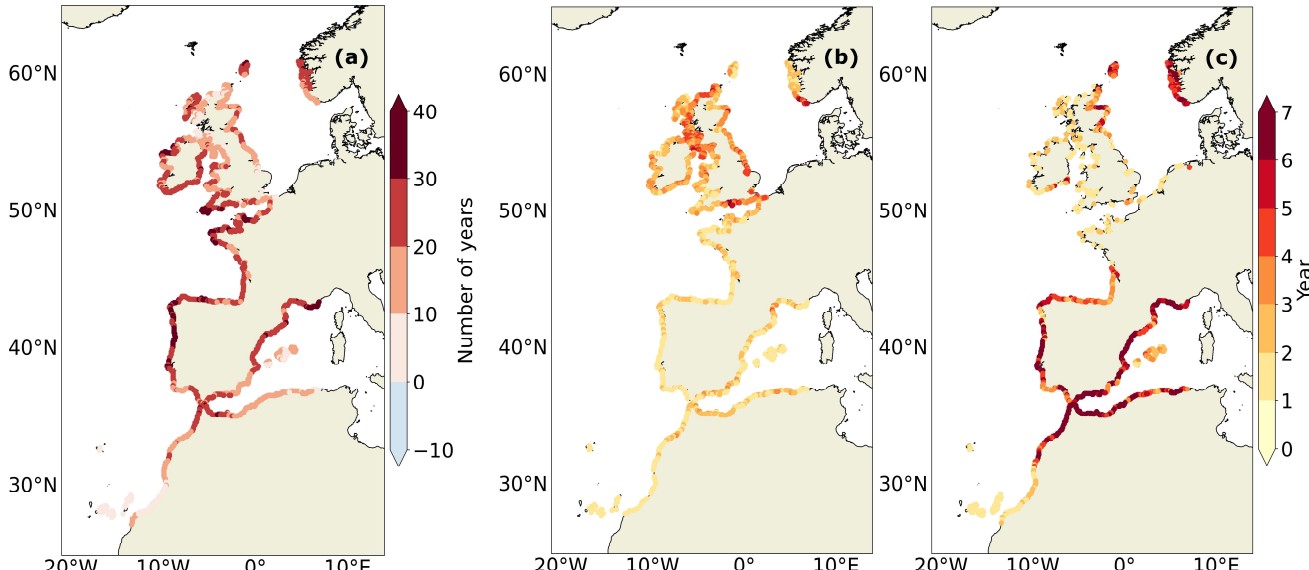

Figure 5: Differences in the year in which the HCE will occur once a year between the SSP1-2.6 and the SSP5-8.5 scenarios for the SWL (Fig. 4b minus Fig. 4a). Only the regions where the confidence intervals of the two scenarios do not overlap are indicated. (b) Confidence intervals for the SSP5-8.5 scenario. (c) Same for the SSP1-2.6 scenario.

## 4.2 Impact of the dynamic approach

The impact of simulating dynamic changes in extremes compared to the usually applied static approach can be assessed with our consistent modelling setup. We start by investigating changes in ESWLs from the static and dynamic approaches. Dynamic changes include changes in all the different simulated components and interactions. This encompasses i. differences in mean SLR between the regional ocean model and the GCM, ii. changes in storm surges and tides (mean and extreme), iii. changes in their interactions, including with mean SLR.

As the uncertainties are larger for the 1-in-100-year event, results are provided here for the 1-in-10-year event. In addition, results for the 1-in-5-year event are included in the Supplementary Materials (Sect. S4), together with results for the 1-in-100-year event. Under the SSP5-8.5 scenario, the largest significant differences between the static and dynamic approaches are in the Mediterranean Sea where the differences in the decennial return level are up to +20%, as well as along the Iberian coast, the Canary Islands and the English Channel (Fig. 6c). Except in the English Channel, these significant differences are mainly due to the differences in the regionalized mean sea level projections (dynamic sea level due to ocean circulations) compared to those of the GCM. For the Mediterranean, the Canary Islands and the southern Iberian coasts, the differences in mean sea level projections between the regional model and the GCM are especially due to bias corrections applied in the regional simulations (Fig. 15a from Chaigneau et al., 2022). Along the northern Iberian coast, the differences are rather attributed to the increased horizontal resolution of the regional model (Fig. 14a from Chaigneau et al., 2022). On the other hand, large negative differences of up to 10% are found in the English Channel and Bristol Channel and are associated with a projected decrease in the mean amplitude of the M2 tidal constituent (Fig. 18 in Chaigneau et al., 2022). Under the SSP1-2.6 scenario, future changes in the different drivers are expected to be of smaller amplitude but so does the increase in mean sea level. In the end, the impact of the dynamic approach (Fig. 6d) is of similar magnitude under the SSP1-2.6 than under the SSP5-8.5 scenario. However, coastal locations exhibiting significant differences are fewer under the SSP1-2.6 scenario (Fig. 6d) owing to the larger confidence intervals (Fig. 5c). Therefore, the use of a dynamic approach should be applied for all scenarios, and not only when high emissions are considered.

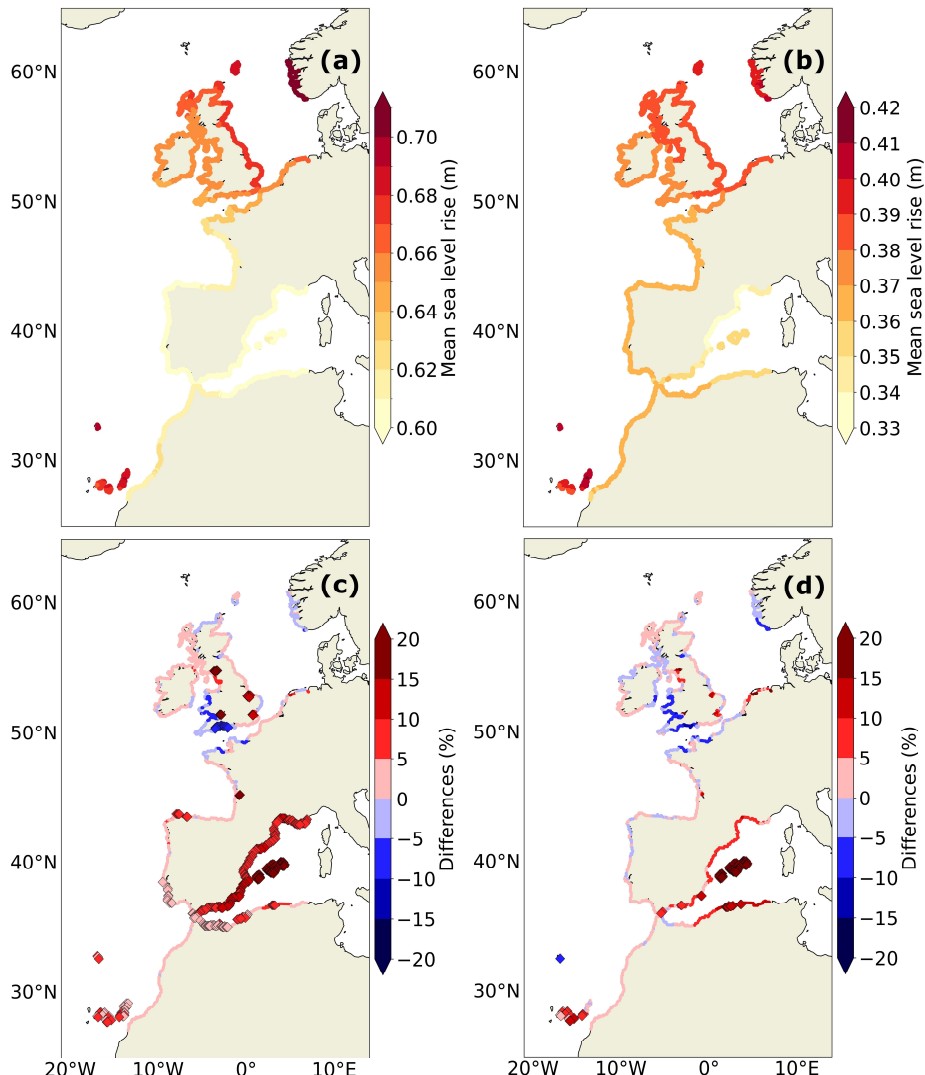

Figure 6: (a) Future changes (2081-2100 minus 1995-2014) in ESWLs using the static approach (i.e. corresponding to the mean SLR shift from the GCM) for the SSP5-8.5 scenario. (b) Same for the SSP1-2.6 scenario. The 1-in-10-year return levels for the historical baseline (1970-2014) are shown in Fig. 3a. (c) Differences (in %) in the projected changes (2081-2100 minus 1995-2014) in the 1-in-10-year ESWLs between the dynamic and static approaches for the SSP5-8.5 scenario. (d) same for the SSP1-2.6 scenario. The diamonds represent the locations where the differences are significant i.e. where the 95% confidence intervals associated with the 1-in-10-year return level calculation for the static and dynamic approaches are disjoint (Fig. 2b).

When including the wave contribution, differences between the static and dynamic approaches additionally reflect changes in wave climate and associated interactions (Fig. 7a). We found less significant impact of dynamic changes in ETWLs than in ESWLs. In fact, over the whole domain, 19% of the coastal points show significant differences for ESWLs (Fig. 6c) and only 5% when waves are included, almost only located in the Mediterranean Sea (Fig. 7a). The isolated effect of dynamic changes in waves on future ESLs is shown in Figure 7b. This effect is generally small over the region, but it tends to reduce future ESLs and therefore to compensate the increase in ESWL amplitude resulting from changes in regionalized mean sea level, storm surges, tides, and interactions (Fig. 6c). This pattern matches the pattern of projected changes in extreme waves from our wave model (Fig. 8b). However, the effect of projected changes in waves on total ESLs remains small compared to the robust decrease of mean and extreme significant wave height and peak period that have been highlighted in several studies along the Atlantic coasts (e.g., Aarnes et al., 2017, Lobeto et al., 2021, Morim et al. 2011, 2023, Chaigneau et al., 2023, Melet et al., 2020) and in Figure 8b.

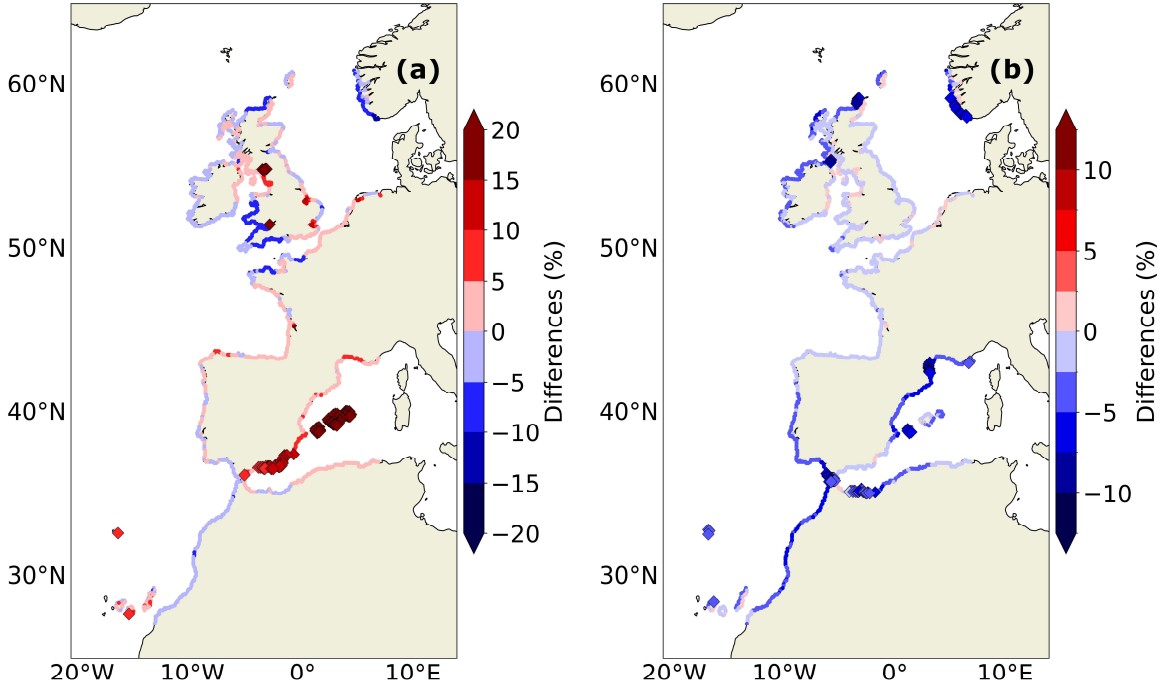

**Figure 7: (a) Differences (in %) in the future changes (2081-2100 minus 1995-2014) in the 1-in-10-year ETWLs between the dynamic and static approaches under the SSP5-8.5 scenario (i.e., same as Fig. 6c but for the ETWLs, therefore including waves). (b) Impact of the inclusion of the dynamic changes in waves on the future ESLs (i.e., differences between Fig. 7a and Fig. 6c). Diamonds represent the locations where the differences between the static and dynamic approaches are significant (Fig. 2b).**

To provide projections of ESLs at large scale considering coastal drivers, some large-scale studies have combined the distribution of the different drivers (e.g., considering their 95[th] percentile separately in Jevrejeva et al., 2023). Compared to these approaches, our modelling setup with a consistent forcing for all drivers of ESLs allow to investigate the co-occurrence of the different contributions to ESLs. For instance, it cannot be inferred that a large reduction in the future wave contribution ($\sqrt{H_s L_p}$) will lead in a reduction of the future amplitude of the ESLs. This is because ESLs are reached due to a combination of different drivers that do or do not necessarily occur at the same time. For instance, local wind forcing can lead to both significant storm surges and extreme waves associated with the wind sea, such as in the Mediterranean and North Seas (Fig. 1). The percentage of the time when extreme events of SWL co-occur with extreme events of waves defined by the wave contribution scaling (Sect. 2.3) is displayed in Figure 8c. Employing wave contribution scaling to explore the timing between the different contributors enables independence from the selected beach slope (Sect. 5). Except in the Mediterranean and southern North Seas dominated by wind sea (Fig. 1), SWL and wave extreme events do not often co-occur. In our domain in general, ESLs are dominated by tides and storm surges. Therefore, the extreme events are rather selected because of the SWL contribution than because of the wave contribution. As extremes in SWL and waves do not often co-occur in the region (apart in the Mediterranean and southern North Seas), the added contribution of waves to ESLs is small. For example, along the Atlantic coasts except the French part, both energetic swells are found (Fig. 8a) and a robust decrease in mean and extreme waves is projected (Fig. 8b) but extremes in SWL and waves rarely co-occur (Fig. 9a). In the southern North Sea, SWL and wave extreme events seem to co-occur frequently (Fig. 9b), but this region is not subject to large projected changes in significant wave height or peak period (Fig. 8b). The only regions where dynamic changes in waves significantly impact changes in ESLs are the Mediterranean Sea, the Norwegian coasts and the Scottish Sea, where both quite large co-occurrence and future changes in wave characteristics occur. This shows that wave future contribution could be neglected in regions where they do not occur at the same time as the dominant contributors, here, tides and storm surges. It would probably be different in regions where the amplitudes of tides and storm surges are smaller and where waves (swell and/or wind sea) dominate the ESLs, as for some tropical coastlines.

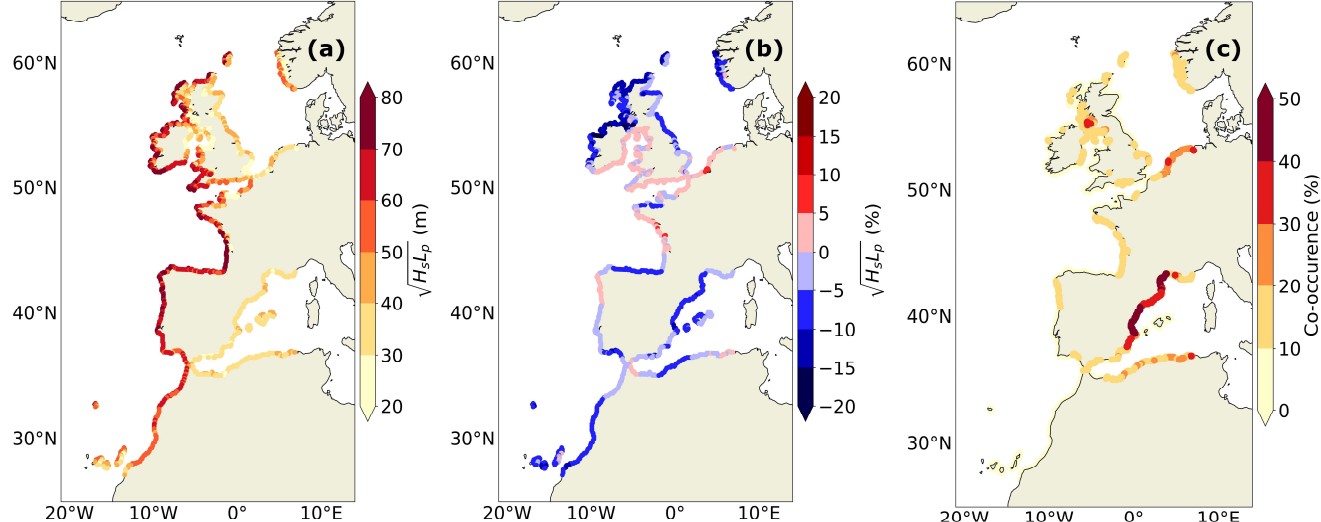

**Figure 8: (a)** The 1-in-10-year wave contribution scaling ($\sqrt{H_sL_p}$, in m) for the 1995-2014 historical period. **(b)** Future changes (2081-2100 minus 1995-2014) in the 1-in-10-year event for the wave contribution scaling ($\sqrt{H_sL_p}$, in %) under the SSP5-8.5 scenario. **(c)** Percentage of time when ESWL and extreme wave events (defined by the wave scaling $\sqrt{H_sL_p}$) co-occur during the 1970-2100 period. It is defined as the ratio between the number of co-occurrences within a 3-day period and the total number of selected ESWL peaks, as illustrated in Figure 9.

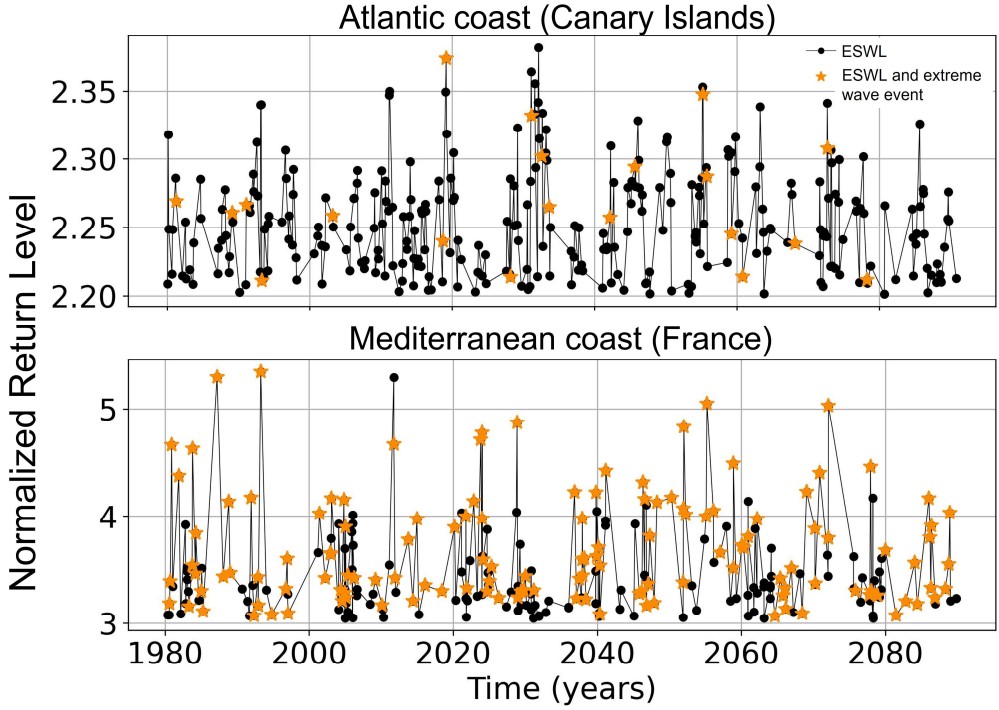

**Figure 9:** Illustration of the selected ESWLs (black dots and yellow stars) as a function of time for the whole period at two different locations, with black dots when ESWLs did not co-occur (i.e. within a 3-day window) with an extreme wave event, and with yellow stars when the ESWL co-occurred (within a 3-day window) with an extreme wave event (defined by the wave scaling $\sqrt{H_sL_p}$). The more yellow stars in the graph, the more ESWL and extreme wave events co-occur. Note that the y-axis shows normalized extremes levels (by the trend and variability, see Sect. 2.4, eq (4)) and not the simulated extremes in meters.

In conclusion, our modelling chain methodology enables the simulation of dynamic changes in ESL drivers. Significant projected changes in the coastal drivers occur, such as for the wave contribution, underscoring the necessity of employing a dynamic approach to generate ESL projections. We found the dynamic approach to be significantly important in the Mediterranean Sea due to the influence of the refined future mean SLR from the regional ocean circulation model. Since mean sea level directly influences changes in ESLs, this emphasizes the importance of downscaling dynamically GCMs, along with potentially bias-correcting their forcings, to resolve ocean circulations and associated mean sea level as accurately as possible. Elsewhere in our region, the relatively small impact of the dynamic approach is attributed to

compensating changes between drivers of ESLs (storm surges, tides, waves, regional mean sea level), that are specific to our region (e.g., decrease in waves and increase in regionalized mean sea level on the Atlantic coast and in the Mediterranean Sea) and driven by a single GCM. For the north-eastern Atlantic coasts, it is also due to differences in timing between extreme coastal sea level contributions. For instance, the robust projected decrease in extreme waves along the eastern Atlantic facade does not significantly impact projected changes in ESLs due to a rare co-occurrence between extreme waves and the dominant processes (i.e. storm surges and tides). In the end, for all these reasons, our findings are in agreement with previous modelling studies using barotropic dynamic approaches (Vousdoukas et al., 2018b; Muis et al., 2020, Jevrejeva et al., 2023) showing that changes in ESLs primarily depend on mean SLR.

## 5 Discussion

### Modelling methodology

The primary focus of our study is on understanding the overall added value of the dynamic approach for ESL projections. Additionally, it would be valuable to explore the relative significance of changes in each contributor to ESLs. Doing so would require conducting dedicated simulations deactivating only one component at a time (tides, storm surges, mean sea level from higher resolution and from corrections of the GCM forcings…), which would be computationally very expensive and was unaffordable for this study.

The results obtained for the dynamic changes in extremes are dependent on the modelling chain that is implemented. For both the ocean and wave models, the representation of the different coastal processes and their interactions is limited by the model horizontal resolution (5-10 km), the corresponding bathymetry and coastline resolution, and by the fact that dry areas are not allowed in the current version of the ocean model. The new version of the NEMO ocean model (v4.2) should improve this limitation by incorporating wetting and drying processes (O'Dea et al., 2020). Additionally, the global mean thermosteric SLR is not included as an input in the ocean model and is instead added a posteriori (Sect. 2.3). Therefore, its impact (with global mean thermosteric SLR projected to be +30 cm by the end of the century under the SSP5-8.5 scenario) on the different components is not accounted for. The impact of waves in the ocean model is also neglected, whereas Bonaduce et al. (2020) highlights a non-negligible contribution of wave-induced processes to sea level, particularly for the extremes. Due to the assumptions made in the ocean model (fixed geoid in particular), some contributions from regional to local sea level variations are also not considered in the projections of extremes, in particular these inducing vertical land motion (such as contemporary GRD effects, glacial isostatic adjustment, sediment deposition or compaction, plate tectonics, local pumping of groundwater and hydrocarbons, and the weight of infrastructure in coastal cities). In some areas it could be interesting to add this contribution a posteriori, for example, for the Baltic Sea and the northern North Sea which are subject to a consequent land elevation due to the glacial isostatic adjustment (Piña-Valdés et al., 2022). Furthermore, considering coupling effects between the ocean, the atmosphere and wind-waves (Lewis et al., 2019) could allow to better resolve coastal processes, notably storm surges that depends on the wind stress. An alternative at a lower computational cost could be the use of a simplified atmosphere such as the atmospheric boundary layer model (ABL, Lemarié et al., 2021) that would allow the ocean feedback on the atmosphere therefore better resolving the winds at the coast.

### Estimation of the wave contribution

In this study, the wave contribution is evaluated based on a generic parameterization (Stockdon et al., 2006), as seen in other climate studies (Melet et al., 2018, 2020; Lambert et al., 2020). This approach appears pragmatic given the wave model resolution of 10 km and the coastal processes that are poorly resolved in the wave model. However, this parameterization comes with notable limitations. It assumes sandy beach conditions, which may not accurately reflect the diverse sediment

types found along many European coastlines, such as rocky shores or mixed sediments. Additionally, the parameterization is designed for deep water conditions, which may not be representative of all coastal points of the domain, as they are not all purely deep water. The model also relies on a prescribed beach slope β, which varies across different coastal areas. Regional estimates of $\beta$ are being developed (Vos et al., 2020) but public estimates of this environmental parameter applicable in empirical formulations are not yet available for the European region. While other studies offer global-scale beach slope information, they typically provide either the nearshore slope (Athanasiou et al., 2019) or the sub-aerial coastal slope (Almar et al., 2021), rather than the foreshore beach slope required in equation (2). Incorporating these values would introduce a regional spatial information that may not be accurate, leading to other type of uncertainties—resulting in either underestimations or overestimations of the wave contribution. Therefore, we opted to maintain a constant representative value of 4% from Melet et al. (2020). Sensitivity analyses were conducted using slopes of 2% and 10% in the Supplementary Materials (Sect. S3). Amplification factors and allowances of ESLs are found to be strongly sensitive to the value of the beach slope. For these reasons, we used here the wave contribution only to derive future changes in the large-scale wave contribution (in %) or to investigate the timing between different contributions, both being independent of the choice of the beach slope. To obtain precise and reliable estimates of coastal wave processes such as wave setup, runup, and total water level for adaptation measures, localized studies are needed (e.g., Serafin et al., 2019). However, our study does not aim to provide such localized estimates.

**Extreme value analysis approach**

The results are also dependent on the choice of the extreme value analysis method (e.g., Wahl et al. 2017), and can be sensitive to the choice of confidence interval calculation method, particularly in unbounded cases such as those found in the northern domain (Scottish coasts, North Sea) and in the Mediterranean Sea when wave contributions are included. However, such cases are not prevalent in our study area. In this study, the shape parameter remains constant over time which is probably not valid for all coastal points of the domain as shown in Supplementary Materials (Sect. S5). Moreover, changes in seasonality and natural variability are not taken into account in the method, whereas Hermans et al. (2022) and Roustan et al. (2022) have reported changes in the seasonal cycle of sea level in the same domain. It would be interesting to compare the results obtained with our (relatively) simplified extreme value analysis method with a more sophisticated method such as a multivariate approach (Arns et al., 2017; Lambert et al., 2020; Marcos et al., 2019; Sayol and Marcos, 2018; Serafin et al., 2017) like the Skew Surge Joint Probability Method where the stochastic surge or wave part is analysed with extreme value models and then combined with the deterministic tidal signal. This would require an estimation of the dependence structure between the different processes/variables to account for the interactions between the different components. Here, the aim was to preserve the simulated dependence between all the extremes by using the direct method. For instance, by applying the direct method based on the whole time series, the extreme value analyses can account for the projected future decrease in tidal amplitude in the English Channel (Fig. 6c).

**Challenge on dynamic changes in extreme sea levels**

Our findings align with previous modeling studies using barotropic dynamic approaches (Jevrejeva et al., 2023; Muis et al., 2020; Vousdoukas et al., 2018), indicating that changes in ESLs primarily depend on mean SLR. This challenges recent research showing that historical trends in storm surges (Reinert et al., 2021; Calafat et al., 2022; Tadesse et al., 2022; Roustan et al., 2022) and tides (Pineau-Guillou et al., 2021) have been comparable in magnitude to historical mean sea level rise trends. However, the conclusions these authors draw from historical trends do not necessarily apply to future trends, which is the main focus of this article. Further research is needed to better understand and quantify dynamic projected changes in all the extreme components, their interactions, and timing (e.g., Melet et al. 2024). Currently, dynamic approaches typically do not account for projected changes in all coastal sea level components (mean sea level, tides, storm

surges, waves, freshwater discharge) or their nonlinear interactions. These approaches often lack resolution to accurately capture the various contributions and their nonlinear interactions, as previously discussed. This can result in a misrepresentation of ESLs and their changes, potentially underestimating the significance of dynamic changes in extremes. Additionally, most studies projecting dynamic changes in extremes rely on small ensembles of model simulations or emission scenarios, similar to our study, due to the high computational cost of simulating all the different components and the limited availability of forcing data (Vousdoukas et al., 2017, 2018; Muis et al., 2020, 2022; Jevrejeva et al., 2023). For instance, global climate models used for driving projections often have relatively low atmospheric resolution, typically around 1° (0.5º in this study), with only a few models being part of the HighResMIP project (0.25º) that better simulate extreme winds responsible for storm surges. Even with a 0.25° resolution, it may still be insufficient to accurately resolve historical and future atmospherically driven contributions, including for instance extra-tropical cyclones in our region. The use of dedicated products such as downscaled atmospheric forcing (e.g., Euro-CORDEX, Outten and Sobolowski, 2021) may offer a promising alternative. Finally, as suggested by Calafat et al. (2022), differences between driving climate models and internal climate variability may also lead to robustness challenges in projecting ESLs. For example, Muis et al. (2022) found little agreement between projected changes in storm surges using different HighResMIP models.

## 6 Conclusions

In this study, regional projections of ESLs are produced along the north-eastern Atlantic coasts taking into account the different sea level contributions such as tides, storm surges, waves, mean sea level and the interactions between these processes. To this aim, regional ocean (3-D baroclinic) and wave (2-D) simulations driven by the same CMIP6 high-resolution GCM are performed over the 1970-2100 period. Under both the SSP5-8.5 and SSP1-2.6 scenarios, large amplifications of ESLs are found all over the region during the 21$^{st}$ century, but the most impacted zone is the southern domain and especially the Mediterranean Sea where the 1-in-100-year sea levels are expected to occur once a year within 20 years. However, the use of a single forcing GCM and a single member does not allow the quantification of the uncertainties of the projected changes in ESLs. Rather, the regional simulations were used to investigate methodological questions related to the production of ESL projections based on regional simulations. More specifically, we assessed the influence of dynamically simulating projected changes in ESLs including the different coastal drivers.

Our dynamic approach accounting for projected changes in the different coastal sea level components (storm surges, tides, waves, regionalized mean sea level) was compared to a static approach where only the mean SLR from the GCM was considered (stationary distribution for other components). The impact of simulating dynamic changes in extremes is found significant in the Mediterranean Sea with differences in the decennial return level of up to +20% compared to the static approach. This is attributed to the refined mean SLR simulated by the regional ocean general circulation model. In general, we observed for the whole domain compensating changes between ESL drivers (storm surges, tides, waves, regional mean sea level) that are influenced by the GCM used as boundary conditions, along with differences in timing among these drivers. This results in future changes in ESLs being primarily driven by mean SLR from the GCM, with coastal contributions having a second order effect, as highlighted in previous research (Vousdoukas et al., 2018b; Muis et al., 2020, Jevrejeva et al., 2023).

In conclusion, the importance of employing a dynamic approach instead of a static one to assess future changes in ESLs is expected to vary across regions. More specifically, the relevance of such an approach relies on the dominating processes and their timing, on the amplitude of projected changes in the GCM forcing used, and on the modelling chain implemented adapted to the features of the region. We found that static projections of ESLs may lead to misleading results in regions

where: i. ESL drivers do not compensate for each other, and ii. extremes in ESL drivers coincide. Furthermore, if these ESL projections rely on mean sea level changes from large-scale models (e.g., GCMs at 100 km resolution), inaccuracies may arise in regions where ocean circulations (mean sea level) are expected to differ significantly from those resolved at larger scale, for instance due to the resolution or bias corrections. The dynamic approach should be considered regardless of the emission scenario: while lower emission scenarios may lead to smaller ESL amplitude changes, this is true for all components, including mean SLR. In specific regions, it would be therefore appropriate to consider dynamic changes in extremes to derive allowances to inform adaptation, as allowances condense all the distribution of sea level projections into a single recommendation (e.g., Howard and Palmer, 2020). This study is situated several steps before the local scale adaptation processes, focusing on identifying regionally key processes or mechanisms to be considered in projections of ESLs.

**Code availability**

The IBI-CCS model is based on the NEMO 3.6 version developed by the NEMO consortium (https://doi.org/10.5281/zenodo.3248739, Madec et al., 2017). All specificities included in the NEMO code (version 3.6) are freely available (NEMO, 2022: https://www.nemo-ocean.eu/).The MFWAM wave model used in this study is based on the wave model WAM, which is freely available at https://github.com/mywave/WAM (last access: 17 July 2023, The Wamdi Group, 1988).

**Data availability**

Data of past and future 1-in-100-year return levels for still and total water levels are available on request. The tide gauge data used for validation are available on the GESLA website (at www.gesla.org). Information on CNRM-CM6-1-HR simulations can be found at https://doi.org/10.22033/ESGF/CMIP6.4067 (CNRM-CM6-1-HR, historical; Voldoire, 2019a), https://doi.org/10.22033/ESGF/CMIP6.4164 (CNRM-CM6-1-HR, piControl; Voldoire, 2019b), https://doi.org/10.22033/ESGF/CMIP6.4225 (CNRM-CM6-1-HR, ssp585; Voldoire, 2019c). The CNRM-CM6-1-HR forcing fields are available on the ESGF website (ESGF, 2022a: historical data, http://esgf-data.dkrz.de/search/cmip6-dkrz/?mip_era=CMIP6&activity_id=CMIP&institution_id=CNRM-CERFACS&source_id=CNRM-CM6-1-HR&experiment_id=historical; ESGF, 2022b: piControl data, http://esgf-data.dkrz.de/search/cmip6-dkrz/?mip_era=CMIP6&activity_id=CMIP&institution_id=CNRM-CERFACS&source_id=CNRM-CM6-1-HR&experiment_id=piControl; ESGF, 2022c: ssp585 data, http://esgf-data.dkrz.de/search/cmip6-dkrz/?mip_era=CMIP6&activity_id=ScenarioMIP&institution_id=CNRM-CERFACS&source_id=CNRM-CM6-1-HR&experiment_id=ssp585 ).

**Author contributions**

AM, AV and AAC designed the study. AAC and GR performed the sea level regional simulations. SLC and LA performed the wave regional simulations. AAC did the analyses of the study, with support of MIA on the extreme value analysis methods. AM, AV, and GR supervised the project. AAC wrote the first draft of the manuscript. All the authors contributed to paper revisions and read and approved the submitted version.

**Competing interests**

The contact author has declared that none of the authors has any competing interests.

**Acknowledgements**

The authors are grateful to Aurélien Ribes for sharing his expertise on the extreme value theory particularly non-stationary methods. The authors are grateful to Lorenzo Mentaschi for providing the code used to perform the extreme value analyses and Jonas Pinault for sharing the required Matlab toolboxes. The authors are also grateful to Jérémy Rohmer for the help and advice on extreme value analyses. We also thank Caldwell et al. (2015) for the central role of the Joint Archive for Sea Level

(JASL)/ University of Hawaii Sea Level Center (UHSLC) in GESLA-3.

**Financial support**

The PhD thesis of Alisée A. Chaigneau was supported by Mercator Ocean International and Météo-France.

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
