# Peer review of "Dynamic Projections of Extreme Sea Levels for western Europe based on Ocean and Wind-wave Modelling"

_EGUsphere, 2024_

## Author Comment (AC1)

11.09.2024

Answer to Referee 1:

The authors wish to thank the anonymous reviewer for his/her comments which greatly helped us to improve the quality of the paper. We are pleased to address the point-by-point answers to your review in blue in the supplement to this comment.

**Additionally, during the review process, we decided to revise the extreme value analysis (EVA) calculation method by removing the constant seasonal component. Recent studies have highlighted changes in the seasonal cycle of sea level within the same domain (Hermans et al., 2022; Roustan et al., 2022), suggesting that the assumption of a constant seasonal cycle may no longer be valid. Therefore, the seasonal term from equation (4) has been removed. The analyses have been re-performed and figures have been updated accordingly, which slightly affects the results. However, this modification does not affect the main text and conclusions of the paper.**

Best regards,

The authors.

**Main comments:**

- One big concern that I have is about the uncertainties in the extreme value analysis, which are crucial since they are used to identify regions where the static and dynamic approaches lead to different results. There is no explanation of how uncertainties in the distribution parameters and resulting return level estimates are derived; is it the Delta Method, or Bootstrapping, or something else? Overall, the uncertainty estimates appear unrealistically small and that would lead to more places being identified where static and dynamic approaches are "different". Looking for example at Figure S1.1 the uncertainties in the 100-year water levels (derived from ONLY 20 years of data) are only a few centimeters in some of the places and maybe 10-20 cm in others for the static approach. They become a bit larger in the dynamic approach because the shape parameter changes signs, but especially for those unbounded distributions the uncertainties are usually very large, particularly when GPD is fitted to short records (as is done here). I don't think this can be correct and would mean that all the conclusion regarding "significant" differences between static and dynamic approaches are corrupted. I am not sure if maybe some uncertainties were ignored (like the ones in the shape parameter) or not propagated properly (as in the Delta Method), but something is off. I know the authors say they use the tool that was already published but that doesn't mean it does the right thing. The reason I give "major revision" is because this is a critical point that may change the results/conclusions.

Thank you for your comment. Uncertainties in the distribution parameters are accounted for. Indeed, the calculation of confidence intervals in the package used for this study (Mentaschi et al., 2016) relies on the Delta Method (asymptotic intervals) which tends to produce narrower and symmetric confidence intervals compared to other methods like the bootstrap method (Caires, 2011). This method has been used to propagate error components related to the uncertainty in estimating the long-term trend and long-term variability (99th percentile) to the error associated with fitting the stationary extreme value distribution, thereby combining both sources of uncertainty. This explanation has been added to the Methods section (Sect. 2.4, end of the "dynamic approach" subsection).

However, the stationary-transform EVA (and associated confidence intervals) were performed using the 131-year period spanning from 1970 to 2100, and not from a 20-year period. The stationary-transform method (Mentaschi et al. 2016) was selected for the EVA as it accounts for the complete, long time series, hence reducing uncertainties in estimating the 1-in-100-year event (tail of the distribution). To transform the non-stationary time series into a stationary one, the long-term trend and variability were calculated using a running mean over a 20-year window (see Mentaschi et al. 2016). Results are thus centered around specific 20-year periods (e.g., 1995-2014, as shown in Figure S1.1). However, the extreme value distribution fit is based on and representative of the entire 131-year time series. Using a longer time series leads to narrower confidence intervals compared to those obtained from a shorter period. This effect is especially noticeable for the 1-in-100-year return levels, as shown in Figure RC1, which compares our 131-year EVA centered on a 20-year period with an analysis based on only 20 years. The legends for all the figures in the paper have been revised to avoid the confusion.

[Figure]

**Figure RC1: Return period curves for ESWL at two of four locations of the Supplementary Materials: The Scottish coasts (unbounded case) and the Brittany coasts in France (bounded case): (blue) computed over the whole 1970-2100 period and centered on the 1995-2014 period, (red) computed only on the 1995-2014 period. Note the different y-axes for the different panels. The dotted lines are the 95% confidence intervals associated with the EVA method applied to compute the ESLs.**

A separate test was performed to highlight the differences between our method and the bootstrap method, which calculates confidence intervals by resampling the peaks with replacement to evaluate the variability and reliability of the estimates. Both methods are not directly comparable because our approach involves normalizing the peaks to account for non-stationarity. For a direct comparison, the uncertainty in estimating non-stationary parameters (i.e., long-term trend and variability) is not considered. Consequently, the resulting curves and intervals differ slightly from those in Figure RC1. In this analysis, the 95% confidence intervals are calculated using a stationary EVA applied to the entire 131-year detrended time series. The time series was detrended using a 20-year running mean. The bootstrap method was performed with 1,000 samples of the peaks obtained from the stationary EVA. The findings indicate that, when using the bootstrap method, the intervals are nearly identical for the bounded case and slightly larger for the upper interval in the unbounded case, but they remain within the same order of magnitude (Fig. RC2). A sentence has been included L401 in the discussion: "The results are also dependent on the choice of the extreme value analysis method (e.g., Wahl et al., 2017), and can be sensitive to the choice of confidence interval calculation method, particularly in unbounded cases such as those found in the northern domain (Scottish coasts, North Sea) and in the Mediterranean Sea when wave contributions are included. However, such cases are not prevalent in our study area."

[Figure]

**Figure RC2: Return period curves for ESWL computed over the 1970-2100 period at two of four locations of the Supplementary Materials: The Scottish coasts (unbounded case) and the Brittany coasts in France (bounded case). The dotted lines are the 95% confidence intervals associated with the EVA method applied to compute the ESLs: with the delta method from Mentaschi et al. (2016) (blue) and bootstrap method (red). Note the different y-axes for the different panels.**

- Calafat et al. (2022; https://doi.org/10.1038/s41586-022-04426-5) showed that trends in storm surges are comparable to trends in MSL at several coastline stretches in Europe and that this has led to pretty large changes in return periods. This clearly challenges the conclusion which is also drawn here in line 370 that "changes in ESL primarily depend on SLR". They also showed that small ensembles cannot capture the full picture of ESL changes. I would like to see some discussion about how the results presented here relate to that.

A new paragraph has been added at the end of the Discussion to address the challenges in capturing dynamic changes in extremes.

**"Challenge on dynamic changes in extremes**

Our findings align with previous modeling studies using barotropic dynamic approaches (Jevrejeva et al., 2023; Muis et al., 2020; Vousdoukas et al., 2018), indicating that changes in ESLs primarily depend on mean SLR. This challenges recent research showing that historical trends in storm surges (Reinert et al., 2021; Calafat et al., 2022; Tadesse et al., 2022; Roustan et al., 2022) and tides (Pineau-Guillou et al., 2021) have been comparable in magnitude to historical mean sea level rise trends. However, the conclusions these authors draw from historical trends do not necessarily apply to future trends, which is the main focus of this article. Further research is needed to better understand and quantify dynamic projected changes in all the extreme components, their interactions, and timing (e.g., Melet et al., 2024). Currently, dynamic approaches typically do not account for projected changes in all coastal sea level components (mean sea level, tides, storm surges, waves, freshwater discharge) or their nonlinear interactions. These approaches often lack resolution to accurately capture the various contributions and their nonlinear interactions, as previously discussed. This can result in a misrepresentation of ESLs and their changes, potentially underestimating the significance of dynamic changes in extremes. Additionally, most studies projecting dynamic changes in extremes rely on small ensembles of model simulations or emission scenarios, similar to our study, due to the high computational cost of simulating all the different components and the limited availability of forcing data (Vousdoukas et al., 2017, 2018; Muis et al., 2020, 2022; Jevrejeva et al., 2023). For instance, global climate models used for driving projections often have relatively low atmospheric resolution, typically around 1° (0.5° in this study), with only a few models being part of the HighResMIP project (0.25°) that

better simulate extreme winds responsible for storm surges. Even with a 0.25° resolution, it may still be insufficient to accurately resolve historical and future atmospherically driven contributions, including for instance extra-tropical cyclones in our region. The use of dedicated products such as downscaled atmospheric forcing (e.g., Euro-CORDEX, Outten and Sobolowski, 2021) may offer a promising alternative. Finally, as suggested by Calafat et al. (2022), differences between driving climate models and internal climate variability may also lead to robustness challenges in projecting ESLs. For example, Muis et al. (2022) found little agreement between projected changes in storm surges using different HighResMIP models."

**Specific comments**

19 "significant" in a statistical sense? If so which significance level? If not ina statistical sense I suggest changing and not using the term in a paper like this where statistical significance is also a big part

Yes, in a statistical sense. The sentence has been changed to: "The impact of simulating dynamic changes in extremes is found statistically significant in the Mediterranean Sea with differences in the decennial return level of up to +20% compared to the static approach."

34-37 what about freshwater discharge?

We did not include freshwater discharge here as the focus is on the ocean contributions to ESLs. But the freshwater input from rivers is indeed taken into account in the ocean model (Chaigneau et al., 2022).

97 return levels

Corrected.

115-120 Would the model resolve changes in (coastal) tides as a result of SLR or is it too coarse?

Yes, the model is accounting for interactions between SLR and coastal tides to some extent. For example, a large projected decrease in the mean amplitude of the M2 tidal constituent is found in the English Channel and Bristol Channel (Fig. 5c here and Fig. 18 in Chaigneau et al. (2022)). However, resolving these interactions may be limited by the fact that the global mean thermosteric sea level rise is not included as an input in NEMO due to the Boussinesq hypothesis and is instead added a posteriori (Sect. 2.3). Therefore, its impact (with global mean thermosteric SLR projected to be +30 cm by the end of the century under the SSP5-8.5 scenario) on the different components, including on tides, is not accounted for. Additionally, the ability to resolve these interactions may be further limited by the model horizontal resolution (5-10 km), the corresponding bathymetry and coastline resolution, and the fact that the NEMO version 3.6 used in this study does not allow for dry areas. The new version of NEMO (4.2) could improve this by incorporating wetting and drying processes (O'Dea et al., 2020). This paragraph has been added in the Discussion L379.

137 after runoff there is a closing bracket but no opening one

Corrected.

153 types

Corrected.

173-201 Somewhere it should be highlighted that a "direct" method is used which fits the GPD to the still water levels which include (often large) deterministic tide signals as opposed to the more appropriate "indirect" methods such as SSJPM where the stochastic surge/wave part is analyzed with the extreme value models and combined with the tides.

This comment is included in the Discussion L405-409 and has been revised: "It would be interesting to compare the results obtained with our simplified extreme value analysis method with a more sophisticated method such as a multivariate approach (Arns et al., 2017; Serafin et al., 2017; Sayol and Marcos, 2018; Marcos et al., 2019; Lambert et al., 2020) like the Skew Surge Joint Probability Method where the stochastic surge or wave part is analyzed with

extreme value models and then combined with the deterministic tidal signal. This would require an estimation of the dependence structure between the different processes/variables to account for the interactions between the different components. Here, the aim was to preserve the simulated dependence between all the extremes by using the direct method. For instance, by applying the direct method based on the whole time series, the extreme value analyses can account for the projected future decrease in tidal amplitude in the English Channel (Fig. 6c)."

198 see my first major comment, it needs to be explained how those confidence levels are derived exactly

See comment above.

Fig. 2 Is that a real example or just (used as) a sketch? If it's a real example it would be good to mention whether it uses the static or dynamic approach.

Yes, it is just a sketch.

242 The German Bight is equally (if not even more) complex than the Dutch coast.

The sentence has been modified: "[…] in complex areas like the German Bight and the Dutch coast."

266 This seems to be quite relevant, what is the reason for not including it in the main manuscript?

The amplification factors including the wave contribution have not been included in the main manuscript because they are sensitive to the chosen constant beach slope value of 4%, which is a limitation of this study. Therefore, as stated in the Discussion, the wave contribution is used in the main manuscript only to derive future changes in the large-scale wave contribution or to investigate the timing between different contributions, both being independent of the choice of beach slope.

Fig. 4 Related to my point about uncertainties it would be interesting here to show the ranges of years including the distribution uncertainties and also see where those ranges overlap between SSPs and where they don't

Thank you for the suggestion. We have added a new figure (Figure 5) that displays the confidence intervals in years for each scenario and highlights the differences between the two scenarios in regions where they do not overlap.

278 This related to my comment above about changes in tides and whether they can actually be resolved along the coast

See comment above.

325 lead to

Thank you, done.

**References**

Arns, A., Dangendorf, S., Jensen, J., Talke, S., Bender, J., and Pattiaratchi, C.: Sea-level rise induced amplification of coastal protection design heights, Sci Rep, 7, 40171, https://doi.org/10.1038/srep40171, 2017.

Caires, S.: EXTREME VALUE ANALYSIS: WAVE DATA, JCOMM Technical Report No. 57, 2011.

Calafat, F. M., Wahl, T., Tadesse, M. G., and Sparrow, S. N.: Trends in Europe storm surge extremes match the rate of sea-level rise, Nature, 603, 841–845, https://doi.org/10.1038/s41586-022-04426-5, 2022.

Chaigneau, A. A., Reffray, G., Voldoire, A., and Melet, A.: IBI-CCS: a regional high-resolution model to simulate sea level in western Europe, Geosci. Model Dev., 15, 2035–2062, https://doi.org/10.5194/gmd-15-2035-2022, 2022.

Hermans, T. H. J., Katsman, C. A., Camargo, C. M. L., Garner, G. G., Kopp, R. E., and Slangen, A. B. A.: The Effect of Wind Stress on Seasonal Sea-Level Change on the Northwestern European Shelf, https://doi.org/10.1175/JCLI-D-21-0636.1, 2022.

Jevrejeva, S., Williams, J., Vousdoukas, M. I., and Jackson, L. P.: Future sea level rise dominates changes in worst case extreme sea levels along the global coastline by 2100, Environ. Res. Lett., 18, 024037, https://doi.org/10.1088/1748-9326/acb504, 2023.

Lambert, E., Rohmer, J., Le Cozannet, G., and Van De Wal, R. S. W.: Adaptation time to magnified flood hazards underestimated when derived from tide gauge records, Environ. Res. Lett., 15, 074015, https://doi.org/10.1088/1748-9326/ab8336, 2020.

Marcos, M., Rohmer, J., Vousdoukas, M. I., Mentaschi, L., Le Cozannet, G., and Amores, A.: Increased Extreme Coastal Water Levels Due to the Combined Action of Storm Surges and Wind Waves, Geophysical Research Letters, 46, 4356–4364, https://doi.org/10.1029/2019GL082599, 2019.

Melet, A., van de Wal, R., Amores, A., Arns, A., Chaigneau, A. A., Dinu, I., Haigh, I. D., Hermans, T. H. J., Lionello, P., Marcos, M., Meier, H. E. M., Meyssignac, B., Palmer, M. D., Reese, R., Simpson, M. J. R., and Slangen, A.B.A.: Sea Level Rise in Europe: Observations and projections, State Planet., 1–106, in press, https://doi.org/10.5194/sp-2023-36, 2024.

Mentaschi, L., Vousdoukas, M., Voukouvalas, E., Sartini, L., Feyen, L., Besio, G., and Alfieri, L.: The transformed-stationary approach: a generic and simplified methodology for non-stationary extreme value analysis, Hydrol. Earth Syst. Sci., 20, 3527–3547, https://doi.org/10.5194/hess-20-3527-2016, 2016.

Muis, S., Apecechea, M. I., Dullaart, J., de Lima Rego, J., Madsen, K. S., Su, J., Yan, K., and Verlaan, M.: A High-Resolution Global Dataset of Extreme Sea Levels, Tides, and Storm Surges, Including Future Projections, Frontiers in Marine Science, 7, 2020.

Muis, S., Aerts, J., Álvarez Antolínez, J. A., Dullaart, J., Duong, T. M., Erikson, L., Haarmsa, R., Irazoqui Apecechea, M., Mengel, M., Le Bars, D., O'Neill, A., Ranasinghe, R., Roberts, M., Verlaan, M., Ward, P. J., and Yan, K.: Global projections of storm surges using high-resolution CMIP6 climate models: validation, projected changes, and methodological challenges, Climatology (Global Change), https://doi.org/10.1002/essoar.10511919.1, 2022.

O'Dea, E., Bell, M. J., Coward, A., and Holt, J.: Implementation and assessment of a flux limiter based wetting and drying scheme in NEMO, Ocean Modelling, 155, 101708, https://doi.org/10.1016/j.ocemod.2020.101708, 2020.

Outten, S. and Sobolowski, S.: Extreme wind projections over Europe from the Euro-CORDEX regional climate models, Weather and Climate Extremes, 33, 100363, https://doi.org/10.1016/j.wace.2021.100363, 2021.

Pineau-Guillou, L., Lazure, P., and Wöppelmann, G.: Large-scale changes of the semidiurnal tide along North Atlantic coasts from 1846 to 2018, Ocean Science, 17, 17–34, https://doi.org/10.5194/os-17-17-2021, 2021.

Reinert, M., Pineau-Guillou, L., Raillard, N., and Chapron, B.: Seasonal Shift in Storm Surges at Brest Revealed by Extreme Value Analysis, JGR Oceans, 126, e2021JC017794, https://doi.org/10.1029/2021JC017794, 2021.

Roustan, J.-B., Pineau-Guillou, L., Chapron, B., Raillard, N., and Reinert, M.: Shift of the storm surge season in Europe due to climate variability, Sci Rep, 12, 8210, https://doi.org/10.1038/s41598-022-12356-5, 2022.

Sayol, J. M. and Marcos, M.: Assessing Flood Risk Under Sea Level Rise and Extreme Sea Levels Scenarios: Application to the Ebro Delta (Spain), JGR Oceans, 123, 794–811, https://doi.org/10.1002/2017JC013355, 2018.

Serafin, K. A., Ruggiero, P., and Stockdon, H. F.: The relative contribution of waves, tides, and nontidal residuals to extreme total water levels on U.S. West Coast sandy beaches, Geophysical Research Letters, 44, 1839–1847, https://doi.org/10.1002/2016GL071020, 2017.

Tadesse, M. G., Wahl, T., Rashid, M. M., Dangendorf, S., Rodríguez-Enríquez, A., and Talke, S. A.: Long-term trends in storm surge climate derived from an ensemble of global surge reconstructions, Sci Rep, 12, 13307, https://doi.org/10.1038/s41598-022-17099-x, 2022.

Vousdoukas, M. I., Mentaschi, L., Voukouvalas, E., Verlaan, M., and Feyen, L.: Extreme sea levels on the rise along Europe's coasts, Earth's Future, 5, 304–323, https://doi.org/10.1002/2016EF000505, 2017.

Vousdoukas, M. I., Mentaschi, L., Voukouvalas, E., Verlaan, M., Jevrejeva, S., Jackson, L. P., and Feyen, L.: Global probabilistic projections of extreme sea levels show intensification of coastal flood hazard, Nat Commun, 9, 2360, https://doi.org/10.1038/s41467-018-04692-w, 2018.

Wahl, T., Haigh, I. D., Nicholls, R. J., Arns, A., Dangendorf, S., Hinkel, J., and Slangen, A. B. A.: Understanding extreme sea levels for broad-scale coastal impact and adaptation analysis, Nat Commun, 8, 16075, https://doi.org/10.1038/ncomms16075, 2017.

---

## Author Comment (AC2)

11.09.2024

Answer to Referee 2:

The authors wish to thank the anonymous reviewer for his/her comments which helped us to improve the quality of the paper. We are pleased to address the point-by-point answers to your review in blue in the supplement to this comment.

**Additionally, during the review process, we decided to revise the extreme value analysis (EVA) calculation method by removing the constant seasonal component. Recent studies have highlighted changes in the seasonal cycle of sea level within the same domain (Hermans et al., 2022; Roustan et al., 2022), suggesting that the assumption of a constant seasonal cycle may no longer be valid. Therefore, the seasonal term from equation (4) has been removed. The analyses have been re-performed and figures have been updated accordingly, which slightly affects the results. However, this modification does not affect the main text and conclusions of the paper.**

Best regards,

The authors.

**Specific comments**

- It is unclear to me how the uncertainties in the return level estimates were estimated. The authors should further elaborate on this point as it is also important for assessing the differences between the static and the dynamic approach.

Thank you for the comment. A sentence has been added to the Methods section (Sect. 2.4) to explain how the confidence intervals are computed: "The calculation of confidence intervals in the package used for this study (Mentaschi et al., 2016) relies on the Delta Method (asymptotic intervals) which tends to produce narrower and symmetric confidence intervals compared to other methods like the bootstrap method (Caires, 2011). This method has been used to propagate error components related to the uncertainty in estimating the long-term trend and long-term variability (99th percentile) to the error associated with fitting the stationary extreme value distribution, thereby combining both sources of uncertainty."

- The authors state that they validated the 1 in 10 year return water level instead of the 1 in 100 year. From an impact/risk assessment perspective however the latter is potentially more important. I therefore believe that the authors should further report on the validation of the 1 in 100 year water level.

As mentioned by the reviewer, the 1-in-100-year level could be potentially more important for impact studies. However, as stated L94-97, we only have 35 years of tide gauge records to validate the model. Therefore, we focused on the 1-in-10-year level in the main text instead of the 1-in-100-year level, as the uncertainties associated with observed estimates of the 1-in-10-year return level are lower (see Tab. S4.1). For example, the differences between the 1-in-100-year ESWLs and ETWLs are significantly smaller than the margin of error computed from the tide gauge estimates, which is not the case for the 1-in-10-year levels. The table S4.1 has been included in the Supplementary Materials to provide the mean validation for different return levels.

| Return level | 1-in-5-year level | 1-in-10-year level | 1-in-20-year level | 1-in-50-year level | 1-in-100-year level |
|---|---|---|---|---|---|
| RMSE ESWLs (m) | 0.40 | 0.43 | 0.46 | 0.50 | 0.53 |
| RMSE ETWLs (m) | 0.32 | 0.33 | 0.35 | 0.38 | 0.40 |
| Mean uncertainty of return levels for tide gauge data (m) | 0.08 | 0.10 | 0.13 | 0.18 | 0.22 |

**Table S4.1:** Comparison of ESL return periods computed from model outputs and tide gauges over 1970-2014: RMSE (in meters), calculated as the root mean squared deviations between modeled return levels and tide gauges return levels (see locations of tide gauges in Fig 3b), for different return periods. Mean uncertainty

calculated for the tide gauge data in meters calculated as the amplitude of the 95% confidence intervals for each return period.

- The authors have employed the empirical Stockdon et al. model for estimating wave contribution. Besides several assumptions associated with the use of this model, the authors have assumed a constant beach slope of 4% (note that Hinkel et al., 2013, used a global value of 2% for estimating erosion). Considering that there are several other datasets (which the authors actually cite) and the fact that one could even use land slope as a proxy, I find the use of a constant slope value a little oversimplistic. The authors have commendably performed a sensitivity analysis to explore the effects of their assumption; nevertheless, if I am not mistaken, the figure in the supplementary material suggests substantial differences (both in absolute values but also in patterns) depending on the chosen value (unless I am misunderstanding something). I think the authors should further elaborate on this point.

The limitations related to the Stockdon et al. (2006) parameterization and particularly the use of a constant beach slope have been expanded and moved from Section 2.3 to a dedicated section in the Discussion:

**"Estimation of the wave contribution**

In this study, the wave contribution is evaluated based on a generic parameterization (Stockdon et al., 2006), as seen in other climate studies (Melet et al., 2018, 2020; Lambert et al., 2020). This approach appears pragmatic given the wave model resolution of 10 km and the coastal processes that are poorly resolved in the wave model. However, this parameterization comes with notable limitations. It assumes sandy beach conditions, which may not accurately reflect the diverse sediment types found along many European coastlines, such as rocky shores or mixed sediments. Additionally, the parameterization is designed for deep water conditions, which may not be representative of all coastal points of the domain, as they are not all purely deep water. The model also relies on a prescribed beach slope $\beta$, which varies across different coastal areas. Regional estimates of $\beta$ are being developed (Vos et al., 2020) but public estimates of this environmental parameter applicable in empirical formulations are not yet available for the European region. While other studies offer global-scale beach slope information, they typically provide either the nearshore slope (Athanasiou et al., 2019) or the sub-aerial coastal slope (Almar et al., 2021), rather than the foreshore beach slope required in equation (2). Incorporating these values would introduce a regional spatial information that may not be accurate, leading to other type of uncertainties—resulting in either underestimations or overestimations of the wave contribution. Therefore, we opted to maintain a constant representative value of 4% from Melet et al. (2020). Sensitivity analyses were conducted using slopes of 2% and 10% in the Supplementary Materials (Sect. S3). Amplification factors and allowances of ESLs are found to be strongly sensitive to the value of the beach slope. For these reasons, we used here the wave contribution only to derive future changes in the large-scale wave contribution (in %) or to investigate the timing between different contributions, both being independent of the choice of the beach slope. To obtain precise and reliable estimates of coastal wave processes such as wave setup, runup, and total water level for adaptation measures, localized studies are needed (e.g., Serafin et al., 2019). However, our study does not aim to provide such localized estimates."

- Line 258 – I find the argument that the replication of the north-south gradient is enough to indicate that the single forcing GCM is "to some extent" representative of the projected changes rather weak. Also, "to some extent" is very vague.

Thank you. We agree with you. This part of the sentence has been deleted.

- The authors conclude that changes in ESL depend on changes in MSL, with coastal contributions having a lesser effect. Considering that some important parts of the coast have not been considered, can the authors really generalise this conclusion based on their results?

A new paragraph has been added at the end of the Discussion to address the challenges in capturing dynamic changes in extremes. Additionally, as explained in the conclusion (L432-440), we cannot conclude that changes in ESLs are only dependent on changes in MSL, as the results are expected to vary by region depending on the dominant processes and their timing, with the magnitude of projected changes in GCM forcing, and with the regional configurations implemented.

**"Challenge on dynamic changes in extremes**

[revised manuscript text omitted]